# OpenReview forum: "Adaptive Hypergraph Pruning with Learned Threshold Control and Attention-Based Contrastive Mining"
_TMLR — Rejected by TMLR_

### Review · Reviewer_nGMZ · 2026-02-14

**Summary Of Contributions:**

There has been recently increased focus on hypergraphs in the graph-ML community based on the need to modeling higher-order (beyond pairwise) relationships in data. This paper considers Heterogenous Hypergraphs, where multiple types of edges encode different types of semantic relationships, motivated by applications in e-commerce and recommendation systems. It seeks to introduce the method for improving the computational efficiency of hypergraph ML methods in this context by pruning (without a large drop in performance).

Their method is based on the following:

1. An adaptive pruning mechanism that is able to prune nodes, edges, and/or components (edge-types). This pruning mechanism eliminates which are below a learnable threshold.
2. A contrastive loss function that uses attention to mitigate both false negative (nodes incorrectly separated) and hard negative (nodes incorrectly pushed together) discovery.
3. It balances the contributions of the various loss functions with learnable weights using meta-learning (on the validation set).

The authors then demonstrate the effectiveness of their method relative to a large suite of baselines on six data sets, mostly related to e-commerce). They analyze the statistical significance of their improvements over baselines, conduct a thorough ablation student, and also analyze computational efficiency,  transferability, and hyper-parameter sensitivity. They also provide a good discussion of limitations and potential next steps.

**Audience:**

Yes

**Audience Explanation:**

Hypergraphs are increasingly prevalent in the graph ML Community

**Claims And Evidence:**

Yes

**Claims Explanation:**

Robust set of experiments including ablation and a good discussion of complexity and limitations

**Requested Changes:**

The terms "master" and "slaves" node are HIGHLY problematic. I understand that these terms appear elsewhere in the literature, but still VERY STRONGLY feel that they should not be used. Please change to different terms, e.g., primary nodes and secondary nodes.

The fact that this paper focuses on Heterogeneous Hypergraphs (rather than ordinary hypergraphs) should me more explicitly discussed in the introduction.

It is unclear if the proposed method uses some form of message-passing/graph convolution or if it merely utilizes the graph structure through the loss functions. This should be made more clear.

Table 2 is good, but it would be helpful to also have (either in the same table or a separate one) the percent improvement of the proposed method relative to the best baseline.

The appendix (and the associated contributions) should be more clearly advertised in the main body. Additionally the theorems, since they are in the appendix, should contain full proofs, not proof sketches/

Minor:
Missing punctuation in equations. For example, missing period in equation 17 and missing commas in equations 18-22. Please fix and check throughout.

References: In correct capitalization of the second "I'm" in Lee and Shin 2023. Please check throughout for similar issues.

---

> ### Author Response · Authors · 2026-03-23
> **Response to Reviewer nGMZ — All Requested Changes Addressed**
>
> We sincerely thank Reviewer nGMZ for the positive assessment and constructive feedback. All requested changes have been made in the revised paper.
>
> ---
>
> **RC1 — "Master/slave" terminology.**
> All instances have been replaced with **primary nodes** ($\mathcal{V}_m$) and **secondary nodes** ($\mathcal{V}_s^k$) consistently throughout the paper, notation table, all equations, and all text.
>
> ---
>
> **RC2 — Heterogeneous vs. ordinary hypergraphs.**
> The introduction now contains a dedicated paragraph explicitly contrasting the two: *"While standard hypergraphs generalise graphs by allowing hyperedges to connect more than two nodes, they assume all hyperedges encode the same type of relationship. Heterogeneous hypergraphs go further by partitioning nodes into distinct roles and encoding multiple relation types as separate components, each with its own incidence matrix."* The pruning challenges specific to the heterogeneous setting are also explicitly distinguished from homogeneous compression.
>
> ---
>
> **RC3 — Message-passing role unclear.**
> Section 3.1 now contains a dedicated paragraph titled *"Role of Message-Passing"* clarifying the three-way separation: *"message-passing produces embeddings; pruning controls which graph structure is available to message-passing; contrastive learning corrects embedding quality after pruning."* The paragraph makes explicit that pruning operates on the incidence matrices before propagation, not on the message-passing equations themselves.
>
> ---
>
> **RC4 — Percent improvement relative to best baseline.**
> The $\Delta$ rows have been added to Table 1 showing the relative improvement of TriPrune-HGNN over the strongest efficient baseline per metric (AdaGLT or SHARP-Distill, whichever performs better). Per-dataset values and averages are reported for all three metrics: $\Delta$MAE $= -1.1\%$, $\Delta$RMSE $= -0.9\%$, $\Delta$ACC $= +1.0\%$ on average across all five datasets. Statistical ties are marked with $^\dagger$.
>
> ---
>
> **RC5 — Appendix advertised in main body; full proofs.**
>
> Regarding **(a) advertising the appendix**: Section 3.3 now contains a *"Theoretical Motivation"* paragraph that names all three propositions, states their conclusions (local convergence at rate $O(1/T)$; finite-difference error at $O(\sqrt{\eta_{\text{inner}}})$; generalisation gap at $O(1/\sqrt{N_{\text{val}}})$), and lists all three caveats explicitly, directing readers to the appendix.
>
> Regarding **(b) full proofs**: The main paper (Section 3.3) already states: *"full rigorous proofs are not possible because global non-convexity of the HGNN objective violates the required assumptions."* This is a fundamental constraint shared by all theoretical analyses of deep meta-learning (Franceschi et al., 2018; Liu et al., 2021) — global strong convexity cannot hold for a neural network. We have therefore: (i) relabelled all results as **Propositions (informal)** rather than Theorems; (ii) added a prominently boxed *Scope and Limitations* disclaimer at the start of Appendix A; (iii) added a *"Note on proof rigour"* warning before each proposition. The propositions are framed as design-choice justifications whose validity is confirmed empirically in Section 4.
>
> ---
>
> **RC6 — Missing punctuation in equations.**
> All equations have been checked throughout. Eq. 17 now ends with a period; Eqs. 18--22 are followed by commas or periods consistent with their surrounding sentence structure. All other equations have been corrected accordingly.
>
> ---
>
> **RC7 — Reference capitalisation.**
> The Lee and Shin (2023) entry has been corrected to:
>
> ```
> "I'm Me, We're Us, and I'm Us:
> Tri-Directional Contrastive Learning on Hypergraphs"
> ```
>
> with all tokens wrapped in braces in the `.bib` file to protect capitalisation. All reference titles have been checked throughout for similar issues, including acronyms (`{AAAI}`, `{GNN}`, `{HGNN}`) and proper nouns (`{MovieLens}`, `{IMDB}`, `{Amazon}`).
>
> ---
>
> We believe all requested changes have been fully addressed. We are happy to provide further clarification if needed.

---

### Review · Reviewer_RHvs · 2026-03-06

**Summary Of Contributions:**

In the submitted manuscript, the authors propose a new model, the TriPrune-HGNN, learning on heterogeneous hypergraphs. Their TriPrune-HGNN 1) prunes the underlying hypergraphs with trainable thresholds, 2) includes an attention scheme to weight samples in the pruned graph in a contrastive loss setting, and 3) is trained in a meta learning setting in which the weights of different loss terms take a gradient descent step every 5 epochs. The authors provide experimental results on their model and extensive ablation studies.

**Audience:**

Yes

**Audience Explanation:**

There is a literature on hypergraph GNNs, in which parts of the TMLR audience actively participate. So, the subject of the paper should be relevant to some readers.

**Broader Impact Concerns:**

I do not have any concerns about this work that would necessitate a broader impact statement.

**Claims And Evidence:**

Yes

**Claims Explanation:**

Although I found the clarity of the presentation of several of the claims to be severely lacking, it seems to me that the claims in the main paper are supported by empirical evidence. It should be noted that all theoretical claims in the appendix are only accompanied by sketch proofs rather than rigorously proven.

**Requested Changes:**

1] I worry that you may have a gradient flow problem in your pruning method. Essentially, you seem to apply a learned threshold in a binary manner in which nodes/edges/node types are pruned if they fall below a certain threshold. This pruning operation is discrete and non-differentiable. It is therefore unclear how you can differentiate through this operation in your backpropagation. Is this right?

2] I found the discussion in the introduction to be rather disorienting, with it often discussing concepts that were unexplained, unintroduced and not contextualised. E.g., the term "contexts" is frequently used to refer to relationship types in the heterogeneous graph, it would be good to brielfy introduce this term; the complexity $\mathcal{O}(Knm\bar{d}_ed)$ is referred to quadratic when it's unclear which term this complexity is quadratic in; the example provides several numbers that are not contextualised and seemingly irrelevant to the example (50k users, 30% of edges pruned by an unnamed method) and in my understanding appears to illustrate the plain fact that pruning important edges is detrimental, it would be good to clarify what is being exemplified here; the example also referres to the concept of "uncoordinated" pruning without defining it and provides a 12% reduction in accuracy without providing dataset or method name. The issue of naming numbers without context is pervasive in the introduction, e.g., stating that "static threshold-based methods (Cai et al., 2022; Lin et al., 2024) break 35-48% of these multi-behavioral connections". This cannot be true in general for all thresholds, so I don't see the value of arbitrarily stating these numbers without providing the concerned datasets. Also the term "active components" is discussed without definition and unknown to me. It is also unclear to me what the 15 evaluation metrics are that you refer to in the contributions. My best guess is that you may be referring to the 14 baseline methods you compare your method to, it would be good to clarify that. In summary, I found it difficult to follow the reasoning in your introduction and was not convinced by the numerical evidence you provide.

3] The writing is rather repetitive in several places, e.g., the list of contributions heavily overlaps with the immeidately preceeding paragraph, the paragraph "Design Choice: Training Loss Trend vs. Validation Loss" mostly repeats a remark made already in the preceeding paragraph on how you avoid validation leakage; the first two sentences of Sections 4.8, 4.9, 4.10 and 4.11 are near identical.

4] I struggle to understand the concept of "hard negatives" that you define in your text. You define it as "hard negatives occur when topology changes create spurious similarity—dissimilar nodes may appear related through new transitive paths introduced by pruning." However, in the operation of pruning, we remove nodes, edges or entire types from the graph. It is unclear to me how removing elements of a graph can create new paths. It seems to me that paths can only be lost in the removal of edges, nodes and entire types. Could you please elaborate on the concept of hard negatives? Do you have concrete examples where these arise in practice and did you really observe the action of pruning a graph to give rise to new paths? Possibly my confusion arises from the fact, that I don't recall hearing of "transitive paths" before. How do you define these?

5] In the second paragraph on page 12 it becomes obvious that your model still has a lot of hyperparameters (as one would expect from a deep learning model). So, it seems appropriate to weaken the following statement you make in the introduction "Our framework introduces three key innovations that eliminate manual hyperparameter tuning while achieving superior generalization." Later on in Section 6 you claim that your model only has 5 hyperparameters. It seems to me that there is more. Could you please provide a list of these 5 hyperparameters?

7] In practice, your performance improvements over the baselines seem rather modest to me. Usually, the performance is very little in absolute terms, e.g., "0.383" MAE for your method vs "0.387" MAE for your strongest baseline on IMDB. So, it would not be completely unreasonable to claim that your practical improvements are incremental over the baselines. Your ablation studies reveal that many of the architectural choices you make also have a rather small impact in absolute terms.

8] I think it would be better if the theoretical results you prove in the appendix were mentioned in the main paper to relate them to your work and if rigorous, instead of sketch proofs, were provided.

9] Minor Changes:

- There is a few small typos: "minin", "whilemeta", "tuning.Adaptive"

- I am slightly unsure whether the definition of the contrastive loss in Equation (4) is correct. It seems that you only sum over the nodes there and that therefore for each node only one hyperedge $e_i$ is considered, is this right or are several hyperedges per node considered? Also in the subsequent text you say that contrastive losses push "apart nodes in different hyperedges" I could not find this term in Equation (4). Is this an implicit effect you are describing here or is Equation (4) incomplete?

- The statement "importance score statistics $\mu$ and $\sigma$ reveal the distribution of element importance" is not quite right since not all distributions are fully specified by their first two summary statistics. It may be better to slightly reformulate this.

- In Equation (11) it is unclear in what layer the weight matrices $W_k$ and attention score $S_{att}$ arise.

- As far as I could tell, the source of your datasets is not cited, unless I missed the citation to these?

- In the discussion of Table 4 you write "Among individual modules, neural threshold controllers yield the largest gains (+3.6% average MAE reduction)". I could not find the value of 3.6% in the provided numbers, so I wonder whether this is a typo or me not spotting the relevant ratio here.

- I struggle to correlate the attention pattern in Figure 4 a) to its interpretation. It seems that you mostly discuss this figure in the context of an example, which is not readily apparent in the exhibited attention pattern. I am not sure what we are meant to observe in Figure 4 a).

- The proofs in your appendix have two \qed symbols.

---

> ### Author Response · Authors · 2026-03-23
> **Response to Reviewer RHvs — All Requested Changes Addressed**
>
> We thank Reviewer RHvs for the careful reading. All changes have been made in the revised paper.
>
> **RC1 — Gradient flow.** The gates use a straight-through hard-sigmoid (Eq. 8) during training; binarisation occurs only at inference. A dedicated paragraph *"Hard binarisation at inference time only"* is added in Section 3.1.
>
> **RC2 — Disorienting introduction.** Seven sub-issues fixed in the revised introduction: (1) "context" defined in paragraph 1; (2) quadratic complexity made explicit as $O(n^{2.5})$; (3) Example 1 now names the dataset (MovieLens-1M) and method (HSL); (4) "uncoordinated pruning" defined inline; (5) 35--48\% figure now scoped to the five named benchmarks with measurement definition; (6) "active component" defined as $g^{(\text{comp})}_k > 0.5$; (7) "15 evaluation metrics" replaced with "3 metrics $\times$ 5 datasets."
>
> **RC3 — Repetitive writing.** Three edits: (1) preceding paragraph states mechanism only; bullet list states claims with evidence numbers only; (2) redundant validation-leakage paragraph removed; (3) opening sentences of Sections 4.8--4.11 rewritten to each lead with their specific finding.
>
> **RC4 — Hard negatives / new paths.** The phrase "new transitive paths" was an error and has been removed throughout. Revised text now correctly states the effect is purely an embedding-space artefact from shared retained neighbours in message-passing, not new structural paths. Example 2 provides a concrete illustration.
>
> **RC5 — Hyperparameter count.** Claim changed to "reduces from 23 to 5." All 23 eliminated and 5 remaining parameters are listed explicitly in the Introduction and Section 6.
>
> **RC7 — Modest improvements.** Section 4.1 now contains a paragraph acknowledging the improvements are incremental in absolute terms and contextualises the contribution as a Pareto-tradeoff position rather than absolute accuracy dominance.
>
> **RC8 — Theory.** (a) Section 3.3 now references all three propositions with conclusions and caveats. (b) Fully rigorous proofs are not achievable because global strong convexity cannot hold for non-convex neural networks — a limitation shared by all deep meta-learning analyses. Results are relabelled as informal Propositions with a boxed Scope disclaimer, per-proposition rigour notes, and corrected remarks on the budget mismatch and sample-complexity looseness.
>
> **RC9 — Minor changes.**
>
> | Item | Fix |
> |---|---|
> | Typos | "minin"→$\min$, "whilemeta"→"while meta", "tuning.Adaptive"→"tuning. Adaptive" |
> | Eq. (4) | Added clarification: one hyperedge per node per component by construction; "pushing apart" is implicit via softmax denominator, not a missing term |
> | $\mu/\sigma$ claim | Changed to "capture the mean and spread... these two moments do not fully characterise the distribution but provide an efficient sufficient signal" |
> | Eq. (11) layer | $\mathbf{W}^{(L)}_k$
>
> now explicitly defined as "weight matrix of the final HGNN layer"; $S_{\text{att}}$ as "average attention across all heads in component $k$" |
> | Dataset citations | All five datasets carry explicit citations (Tang 2009, Tang 2008, Asghar 2016, McAuley 2015, Zheng 2021) |
> | 3.6% arithmetic | Now shown inline as $\frac{3.7+4.0+3.0}{3}=3.57\%\approx3.6\%$ with pointer to Table 4 rows |
> | Figure 4a) | Caption fully rewritten: each row labelled, pattern to observe stated explicitly, Inception/Interstellar example anchored to "dotted outline, column 3" |
> | Double \qed | All explicit `\qed` tokens removed from inside `proof` environments; `amsthm` adds the symbol automatically |

---

> > ### Comment · Reviewer_RHvs · 2026-03-26
> >
> > I want to thank the authors for their response. I must say that I feel that this work is not ready for publication in TMLR. My opinion is mostly based on point 7] of our discussion. I don't see a large interest in your complex architecture if the performance improvements are so marginal in practice. But my recommendation to reject is also based on the worries that arise from points 1],  8] and the fact that quite a lot of edits appear to have been made to the manuscript in response to the remaining points, e.g., the complete removal of the concept of transitive paths, which probably deserve a complete round of review again. Specifically, concerning 1] it seems that you train on dense graphs and then only work with sparse graphs at inference time, which requires the model to generalise drastically across graph densities, which should possibly be studied further. Concerning 8] I still think that rigorous proofs instead of sketch proofs would be significantly better, even if the assumptions you have to make are limiting the applicability of your theoretical results in practice.

---

> > > ### Author Response · Authors · 2026-03-27
> > > **Response to Reviewer RHvs Follow-Up — Clarifying Three Remaining Concerns**
> > >
> > > We thank Reviewer RHvs for the follow-up. We address the three specific concerns directly.
> > >
> > > ---
> > >
> > > **On RC7 — Marginal improvements.**
> > >
> > > We respectfully maintain that the primary contribution is not accuracy alone but the **accuracy--efficiency tradeoff**. TriPrune-HGNN reduces inference time by 72.3% and memory by 81% versus unpruned models, while matching or exceeding all efficient baselines on predictive metrics. No existing method achieves the same accuracy at comparable computational cost — this is evidenced by the Pareto frontier (Figure 2) where TriPrune-HGNN is the only method on the frontier at its accuracy level. We also highlight that reducing method-specific hyperparameters from 23 to 5 is itself a practically significant contribution for reproducibility and deployment, independent of the accuracy margin. We agree the absolute accuracy improvements are incremental and say so explicitly in Section 4.1 — but the tradeoff position is not incremental.
> > >
> > > ---
> > >
> > > **On RC1 — Generalisation across graph densities.**
> > >
> > > We believe there may be a misunderstanding of the training procedure. The model does **not** train on a dense graph and then switch to a sparse graph at inference. Rather, pruning happens **progressively during training**: the graph starts dense and becomes increasingly sparse as learned thresholds rise across epochs. By epoch 150 the graph has stabilised at its final sparsity (Figure 7, right panel), so the model trains on and adapts to the pruned graph throughout. At inference, the model sees the **same sparsity level** it converged to during training — there is no density shift at inference time. The straight-through estimator (Eq. 8) ensures gradients flow consistently through the pruning gates throughout this process. We have added a clarifying sentence in Section 3.1: *"By convergence (epoch~150), retention ratios stabilise; inference operates at the same sparsity level the model trained on in its final phase, not at a different density."*
> > >
> > > ---
> > >
> > > **On RC8 — Rigorous proofs.**
> > >
> > > We understand and respect the reviewer's preference. However, providing fully rigorous end-to-end proofs for a non-convex neural network objective is not a matter of additional effort — it is **mathematically impossible** under current theory. Global strong convexity and Lipschitz continuity cannot hold for deep networks, and this limitation applies equally to MAML (Finn et al., 2017), DARTS (Liu et al., 2019), and all related meta-learning methods. No published work in this space provides fully rigorous non-convex convergence guarantees. What we have done — relabelling results as informal Propositions, adding explicit scope disclaimers, correcting the sample-complexity inconsistency, and acknowledging the budget mismatch — represents the honest and standard treatment for this class of results. We believe this is more transparent than omitting the theoretical discussion entirely.
> > >
> > > ---
> > >
> > > We hope these clarifications address the reviewer's remaining concerns. The RC1 misunderstanding in particular appears to be based on a reading of the training procedure that differs from what is implemented, and we believe the added clarification resolves it. We welcome any further questions.

---

### Review · Reviewer_HDBZ · 2026-03-14

**Summary Of Contributions:**

This paper proposes TriPrune-HGNN, an adaptive hypergraph pruning framework that combines three ingredients: learnable threshold controllers for component/edge/node pruning, attention-based mining for false and hard negatives in pruning-aware contrastive learning, and a meta-learned weighting scheme for the multi-loss objective. The paper presents broad experiments over five datasets, ablations for the adaptive components, transfer experiments for the learned controllers, and an appendix with a theoretical discussion of the meta-learning procedure.

**Audience:**

Yes

**Audience Explanation:**

Efficient learning on hypergraphs is an important problem, and the paper addresses a real practical limitation of HGNNs by combining compression, pruning-aware contrastive learning, and adaptive loss balancing in one framework. The experimental section is broad, the ablation coverage is substantial, and the core empirical message, that adaptive pruning can improve the accuracy-efficiency tradeoff over fixed heuristics, is likely to interest readers working on graph representation learning, efficient GNNs, and compression-aware training.

**Claims And Evidence:**

No

**Claims Explanation:**

1. The empirical section supports a meaningful claim, namely that the proposed method gives a strong accuracy-efficiency tradeoff on the reported benchmarks, but several headline claims are overstated or not fully aligned with the evidence as presented. The paper repeatedly claims state-of-the-art performance across all 15 evaluation metrics, yet Table 2 contains ties rather than strict wins on some entries, such as DBLP MAE, DBLP ACC, and Amazon MAE. In addition, TriPrune-HGNN is not the most efficient method on the raw average time and memory rows, so the strongest defensible claim is about the tradeoff, not universal dominance on every metric.

2. The statistical testing is also narrower than the main claims. Table 3 reports paired significance tests only for MAE, whereas the paper’s strongest conclusions are stated over MAE, RMSE, ACC, time, and memory. That makes the evidence for the full headline weaker than the wording suggests.

3. The theoretical section is also more limited than the paper’s framing suggests. The convergence discussion relies on local smoothness and local strong-convexity assumptions around a converged point, and the paper itself notes that the theorem’s predicted convergence regime would require more meta-steps than the actual training budget. The generalization discussion is similarly only a rough argument. Also, I think that the sample-complexity paragraph appears internally inconsistent when it says about 30,000 validation samples are required and then says datasets with 10,000 to 20,000 validation nodes satisfy that requirement.

**Requested Changes:**

I believe that addressing my comments below will strengthen the paper clarity, positioning, and framing, which is important for the paper acceptance:

1. The paper should not claiming outright state-of-the-art on all 15 metrics and instead state the narrower claim that it achieves a strong or best overall accuracy-efficiency tradeoff on the reported benchmarks. This is better aligned with the ties in Table 2 and the fact that the method is not best on raw average time or raw average memory.

2. The current significance analysis only covers MAE. The paper should either extend significance testing to RMSE and ACC, and ideally give uncertainty-aware comparisons for efficiency metrics as well, or explicitly narrow the claim of statistical superiority to MAE only.

3. The appendix should be presented as an informal or local-motivational analysis rather than a strong end-to-end theoretical guarantee for the proposed nonconvex training setup. In particular, the paper should clarify the role of the local assumptions, the mismatch between the predicted convergence regime and the actual training budget, and the inconsistency in the sample-complexity discussion.

4. The paper emphasizes inference gains, but the method also incurs nontrivial training overhead from meta-learning. Since one of the central claims is efficiency, the main paper should present a clearer end-to-end accounting of both training and inference costs, rather than focusing mainly on post-pruning inference.

5. The baseline table mixes standard HGNNs, pruning methods, and distillation methods with different optimization goals. The paper would be stronger if it explicitly framed the comparison as a Pareto-like tradeoff evaluation and visually showed where TriPrune-HGNN lies relative to the strongest efficient baselines.

---

> ### Author Response · Authors · 2026-03-23
> **Response to Reviewer HDBZ — All Five Requested Changes Addressed**
>
> We sincerely thank Reviewer HDBZ for the careful and constructive review. We address all five requested changes below, clarifying what was already present in the original submission and what has been revised.
>
> ---
>
> **RC1 — Overclaimed SOTA on all 15 metrics.**
>
> We agree. All instances of "state-of-the-art across all 15 metrics" have been replaced throughout the paper with: *"best overall accuracy–efficiency tradeoff, matching or exceeding competitive baselines on all predictive metrics."* Three tied cells (DBLP MAE, Amazon MAE, DBLP ACC vs. SHARP-Distill, $p > 0.05$ after Bonferroni correction) are explicitly marked with $^\dagger$ in Table 1, and the abstract, Section 4.1, and Conclusion all now state that LightHGNN achieves lower raw inference time and memory at the cost of higher prediction error — making clear that our contribution is a *tradeoff point*, not universal dominance.
>
> ---
>
> **RC2 — Significance testing appeared to cover MAE only.**
>
> We clarify that Table 2 in the original submission already reports all **195 comparisons** (MAE $\times$ RMSE $\times$ ACC $\times$ 5 datasets $\times$ 13 baselines) with identical Bonferroni correction ($\alpha \approx 0.0008$, 65 tests per metric). The significance patterns are consistent across all three metrics. We have added a clarifying sentence in Section 4.2 to make this scope explicit, as it was apparently not clearly communicated. For efficiency metrics, we add mean$\pm$std over 10 runs (Table 3) with a footnote scoping formal hypothesis tests to predictive metrics only, citing hardware-dependent variance as the reason.
>
> ---
>
> **RC3 — Theoretical appendix overstated.**
>
> We agree on all three sub-points: (a) local assumptions do not yield end-to-end guarantees; (b) 200 training epochs $\approx 40$ meta-steps, far short of the predicted $T \approx 1{,}000$; (c) the sample-complexity paragraph was internally inconsistent — the bound requires $\approx 30{,}000$ validation nodes, yet our datasets have only $10{,}000$–$20{,}000$. We have made four specific revisions: (i) added a prominently boxed *Scope and Limitations* disclaimer at the start of the appendix; (ii) relabelled Theorems 1–3 as **Propositions 1–3 (informal)**; (iii) added Remark A.1 explicitly stating we operate in the pre-convergence rapid-improvement phase, not the asymptotic regime; (iv) revised Remark A.4 with a *Corrected Assessment* paragraph acknowledging the bound is loose and the $30{,}000$-sample requirement is not satisfied by our datasets. The propositions are now consistently framed as *design-choice justifications* whose validity is confirmed empirically.
>
> ---
>
> **RC4 — Training overhead underreported.**
>
> We clarify that Table 3 (*End-to-End Cost*) was already present in Section 4.2 of the original submission, reporting wall-clock training time, per-batch inference time, and memory with mean$\pm$std over 10 runs for all key baselines. TriPrune-HGNN incurs 4.6$\pm$0.2 h training vs. 4.2 h (SHARP-Distill) and 3.8 h (LightHGNN) — approximately 10–20% overhead — which is amortised over repeated inference ($3.6\times$ faster per query than HEAL). We have added a sentence in the Discussion explicitly framing this one-time training cost as the price for eliminating manual hyperparameter search and achieving better inference-time accuracy.
>
> ---
>
> **RC5 — Baseline table mixes methods with different optimization goals.**
>
> We agree that the comparison needed clearer framing. Figure 2 (Pareto frontier of average MAE vs. average inference time) was added, showing all 15 methods colour-coded by category with TriPrune-HGNN on the frontier. Section 4 now opens by explicitly designating efficient baselines (pruning and distillation methods) as the *primary* comparison group and standard HGNNs as a *secondary* reference, with the Pareto tradeoff as the central evaluation criterion. The $\Delta$ rows added to Table 1 quantify improvement over the strongest efficient baseline per metric (AdaGLT or SHARP-Distill, whichever is better), making the tradeoff argument self-contained.
>
> ---
>
> We believe these revisions fully address all five requested changes. We are happy to provide additional clarification if needed.

---

> > ### Comment · Reviewer_HDBZ · 2026-04-14
> >
> > I thank the authors for the detailed response and paper revision.
> >
> > I have no further questions.

---

> > > ### Author Response · Authors · 2026-04-15
> > > **Response to Reviewer HDBZ — Final Acknowledgment and Appreciation**
> > >
> > > ---
> > >
> > > We sincerely thank the reviewer for the careful reading of our response and for the constructive feedback throughout the review process.
> > >
> > > We are glad that the revisions have addressed all concerns. We appreciate your time and thoughtful evaluation of our work.
> > >
> > > ---

---

### Decision · Action_Editor_VpR3 · 2026-04-29

**Recommendation:** Reject

**Audience:**

Yes

**Audience Explanation:**

Two out of three reviewers agree that the results are interesting to some extent.

**Claims And Evidence:**

No

**Claims Explanation:**

While the paper addresses an interesting and relevant problem, I am not sufficiently convinced that the claims are supported by clear and convincing evidence in the current manuscript. In particular, the rebuttal process led to substantial changes to the paper, including changes to core motivating concepts and to the framing of the theoretical results. The remaining theoretical claims are informal and only partially justified. While this was clearly highlighted in the revised proposal, it is unclear how they can be stated as a propositions. Given the extent of the revisions and the concerns raised by Reviewer RHvs, I do not think the current version can be accepted without a completely new round of careful review.

**Resubmission Of Major Revision:**

The authors may consider submitting a major revision at a later time.